# Biodiversity and Distribution of *Reticulitermes* in the Southeastern USA

**DOI:** 10.3390/insects13070565

**Published:** 2022-06-22

**Authors:** Allison Johnson, Brian T. Forschler

**Affiliations:** Department of Entomology, University of Georgia, Athens, GA 30602, USA; bfor@uga.edu

**Keywords:** subterranean termite, mtDNA markers, integrative taxonomy, species distribution

## Abstract

**Simple Summary:**

Describing global biodiversity involves identifying species and describing their distributions. The subterranean termite genus *Reticulitermes* represents an important group of wood-destroying organisms; however, little is known about their species-specific distribution across the three northern hemisphere continents where they are endemic. We combined several taxonomic methods to identify the species of over 4000 specimens in the first statewide survey of subterranean termites from Georgia, USA. The area surveyed, 153,900 km^2^, represents eco-regions typical of most of the southeast and eastern seaboard of the United States. There were three species, *R. flavipes,* *R. virginicus*, and *R. nelsonae,* found throughout Georgia. *R. malletei* was predominantly collected in the northern Piedmont soil province, while *R. hageni*, the least encountered species, was not collected from South Georgia. Our findings support the need for a taxonomic revision of the genus *Reticulitermes*, agreement on an appropriate integrated taxonomic approach for species determination, and should stimulate future research on diverse topics such as biodiversity, monitoring for these structural pests, and identifying their role in forest ecosystems.

**Abstract:**

*Reticulitermes* subterranean termites are widely distributed ecosystem engineers and structural pests, yet describing their species distribution worldwide or regionally has been hindered by taxonomic uncertainties. Morphological plasticity confounds the use of taxonomic keys, while recent species descriptions and molecular techniques lacking taxonomic support have caused a muddle in interpreting the literature on *Reticulitermes* species distributions. We employed an integrative taxonomic approach combining behavioral, morphological, and molecular techniques to identify 4371 *Reticulitermes* samples to species. Five *Reticulitermes* species were collected from wood-on-ground at 1570 sites covering 153,900 km^2^ in the state of Georgia, USA. Three species were collected throughout Georgia, with *R. flavipes* identified from every one of the 159 counties. *R. nelsonae* was the second most frequently collected species, found in 128 counties, with *R. virginicus* third with 122. Two species had distributions confined to the northern part of the state. *R. malletei* was collected from 73 counties, while the least collected species, *R. hageni,* was found in 16. Results show that the most recently described species (*R. nelsonae*, 2012) is widely distributed and the second-most frequently encountered termite, representing 23% of all samples. The invasive species *R. flavipes* represented half of all the samples collected, while *R. hageni,* the least at less than 1%. A search of GenBank identified a number of accessions mismatched to a species designation resulting in the literature under-reporting the biodiversity of the genus. We, therefore, outline a path to standardize methods for species identification using an integrated taxonomic approach with appropriate barcodes for consistent identification across research teams worldwide. The data also illuminate new opportunities to examine questions related to the ecology, evolution, dispersal, and resource partitioning behaviors of these sympatric species across distinct geographical regions.

## 1. Introduction

Biodiversity is a term used to describe the variety of living organisms on Earth that is founded on the nomenclature used to identify an animal life form to a species designation. A number of broad issues constrain delineating species, not the least of which are long-standing, global concerns affecting all biological systems, including the taxonomic impediment [1], taxonomic wrong-headiness [2], and lack of consensus on appropriate integrative methods for species identification [3,4,5,6,7] all contingent on adherence to the internationally agreed codes of rules for animals outlined in International Code of Zoological Nomenclature (ICZN) [8]. Measuring insect biodiversity is a critical component of understanding the impact of anthropogenic changes affecting life on Earth, as exemplified by discussions surrounding the so-called insect apocalypse [9,10]. The acknowledged diversity of insects, however, requires using a common system for identifying species that, if not consistent across studies, often hinders comparing results on any number of topics, including insect distributions [11].

The renowned ecological and economic impact of subterranean termites [12,13,14,15] has not translated into a concerted effort to understand the global distribution of *Reticulitermes* species across their Holarctic range. Subterranean termite risk maps of the USA place Georgia, and surrounding states, in the middle of a ‘very-high hazard’ termite belt [13,16,17,18]. Acknowledging the dense populations in the region has, however, translated into few concerted appraisals of *Reticulitermes* distributions in that same area. Impediments for synthesizing termite distribution data from the literature include the variety of techniques used to collect *Reticulitermes,* such as examining wood on ground, in-ground surveillance/monitoring devices, light traps, collections from Pest Management Professionals (PMPs) or property owners, and museum collections (Appendix A). Discerning *Reticulitermes* distributions in the southeastern USA from the literature is also complicated by two species descriptions in the last thirty years [19,20], meaning that surveys completed before 2012 should be interpreted with discretion. Nelson et al. [21] eloquently, more than a decade ago, detailed the main obstacles to *Reticulitermes* species identification while echoing the call for a taxonomic revision of the genus that goes back over 50 years [22].

The usefulness of taxonomic keys for *Reticulitermes* species identification is confounded by morphological plasticity and resulting range of phenotypes [21,23,24,25,26,27]. Molecular markers have numerous advantages for obtaining consistent across-research program species identification, yet markers used for *Reticulitermes* identification in the southeastern USA have been published with questionable, if any, taxonomic support, especially prior to 2012 rending those designations suspect [21,28,29,30]. An integrative taxonomic approach [ITA] using molecular markers must include reference sequences associated with specimens identified by morphology, behavioral attributes, or chemical signatures to interpret phylogenies and increase the confidence in species designations [3,31,32].

The objective of this study was to identify specimens using an ITA-validated mtDNA marker applied to 4371 samples collected from wood-on-ground (WoG) at 1570 wildland sites over the entire state of Georgia, USA, to illuminate species distributions and proportions. Results are reported for five described species, *Reticulitermes flavipes* (Kollar) 1837 (Rf), *R. virginicus* (Banks) 1907 (Rv), *R. hageni* Banks 1920 (Rh), *R. malletei* Clément et al. 1986 (Rm), *R. nelsonae* Lim and Forschler 2012 (Rn) and discussed in relation to the literature on the taxonomy, ecology, evolution, management, and future research directions with these pestiferous ecosystem engineers.

## 2. Materials and Methods

### 2.1. Sample Area

This survey sampled over 153,900 km^2^ or the entire state of Georgia, USA, and with 1570 sampling sites provided, on average, a sample for every 98 km^2^ between 30–35° N, 85–81° W. Georgia includes five soil provinces following the USDA-NRCS classification [33], with elevations ranging from 0–1400 m [34]. The two largest geographical regions, Piedmont and Coastal Plain, are separated by the Fall Line, a prehistoric shoreline during the Cretaceous period (66–140 mya) (Figure 1). The Fall Line extends from Alabama to New Jersey, separating the Piedmont from Coastal Plains in those southeastern states [35]. The land south of the Fall Line was subject to cycles of rising and falling sea levels during the quaternary period as the shoreline receded to present-day levels exposing the Coastal Plains and Atlantic Coastal Plains Soil Provinces between 20–5 million years ago [36].

### 2.2. Sample Collection

Samples were collected by a single collector from January 2015 to March 2017 from wood on ground (WoG) off of state highways and roads in Georgia, USA. Sampling routes were chosen based on access by roadways that provided coverage of accessible areas of every county (n = 159) in the State. Sampling was conducted within a 100 m radius of a GPS-recorded point termed a site, and there were 5–17 sites in each county. Sampling was conducted by locating all wood on ground (WoG) such as stumps, logs, or coarse woody debris and opening the WoG using a hatchet and chisel to collect all termites into vials of 100% ETOH. The number of samples per site ranged from 1 to 7. The survey included 4371 samples from 1570 sites (1 site per 98 km^2^) across all counties (n = 159) in the State of Georgia, USA.

### 2.3. Sample Processing

One termite from each sample was processed by extracting Genomic DNA using an adapted protocol (Appendix A) from the Wizard^®^ Genomic DNA Purification Kit (Promega, Fitchburg, WI, USA). Resuspended DNA was amplified using a 10 μL volume PCR protocol (Appendix A) with published primers Modified A-t Leu [37] and B-t Lys [38]. PCR conditions were set in an Eppendorf Mastercycler X50s (Hamburg, Germany) at 95 °C for 2 min (1 cycle), 95 °C for 15 s, 53 °C for 15 s, and 68 °C for 45 s (35 cycles), followed by an extension of 68 °C for 5 min (1 cycle). The PCR product was purified using enzymatic digestion of 0.2 μL Exonuclease I (New England Biolabs, Ipswich, MA, USA) and 0.2 μL Calf Intestinal Alkaline Phosphatase (Quick CIP; New England Biolabs, Ipswich, MA, USA) per reaction. PCR purification conditions were set in an Eppendorf Mastercycler X50 s (Hamburg, Germany) at 37 °C for 15 min (1 cycle) and 85 °C for 15 min (1 cycle). The product was diluted 1:1 with Type 1 Ultrapure Water (Direct-Q 3 UV; Millipore Sigma, Burlington, MA, USA). Purified PCR product was sent to Eurofins Genomics (Eurofins, Louisville, KY, USA) for 96-Well Microplate Sanger sequencing. 

### 2.4. Sample Identification

A reference sequence (Appendix A) was an mtDNA haplotype attributed to a species designation from a sample, identified using morphology and flight phenology [27]. The reference sequences were included in a GTR + G + I model maximum likelihood tree with 1000 bootstrap replicates in IQ-TREE v2.6.12 to identify species-specific clades [39]. A published *Coptotermes formosanus* sequence (Ref no. AY683218) from GenBank was utilized as the extant group. Species designations for all haplotypes were inferred based on alignment with the aforementioned reference sequences in trees annotated and visualized in FigTree v1.4 (http://tree.bio.ed.ac.uk/software/figtree/: accessed on 12 May 2022). 

### 2.5. GIS Visualization

All site coordinates were recorded with a GPS unit and compiled in an open-source geospatial information system (GIS) with identification attributes. *Reticulitermes* distribution across Georgia’s geographic features was visualized in QGIS Geographic Information System v 3.22.3 (QGIS Association; http://www.qgis.org: 28 March 2022). Features included a base map from the Georgia Association of Regional Commissions and 2016 Major Land Resource Areas soil survey data.

### 2.6. Spatial Data Analysis

Clark and Evans aggregation indices and Kernel Density Estimation maps were analyzed and developed in R 4.1.2 (The R Foundation for Statistical Computing, Vienna, Austria) using maptools, rgdal v1.5-28, sf v1.0-7, sp, and spatstat v2.3-3 packages [40,41,42,43,44]. Edge effects for Clark and Evans aggregation indices were corrected using the state border of Georgia as a guard buffer. Hill Numbers [45] for species richness and diversity were calculated for species at sites from the entire state as well as above and below the Fall Line [46]. Exponential Shannon entropy indices (H1) were calculated using:e^(−Σ*pi* × *ln*(*pi*))(1)
where *pi* is the proportion of species collected from sites [47]. Hill Number (H2) was calculated using: 1/(**Σ**n(n − 1)/(N(N − 1)(2)
where N is the total number of species collected from sites and n is the total for each species collected from sites [48].

## 3. Results

### 3.1. Survey

We collected termites from all 159 counties at 1570 wildland sites illuminating the ubiquity of *Reticulitermes* as part of the soil macroarthropod fauna in Georgia. There were 78 sites (5%) where we collected three species, 643 sites (41%) had two species, 851 (54%) sites provided one species, and there were no sites where we identified four or more species. On a broader scale, 100% of the counties (P = 159) supplied at least two species, including five species from 7 counties, four species from 32 counties, three species from 98 counties, and two species in 22 counties (Appendix A). *R. flavipes*, *R. virginicus,* and *R. nelsonae* were collected throughout the state while *R. malletei* and *R. hageni* were found only in counties north of the 32nd parallel roughly approximating the geologic featured called the Fall Line (Figure 1).

*Reticulitermes flavipes* was the most frequently encountered termite, identified from every county in Georgia (P = 159), collected at 1131 sites (72%), and represented in 2260 samples (52%) that were distributed almost equitably north (53%) and south (51%) of the Fall Line (Figure 2, Table 1). *Reticulitermes nelsonae* was the second most common species collected statewide at 128 counties (81%), 559 sites (36%), and 1031 samples (23%) (Table 1). *R. nelsonae* was found in a higher proportion of counties below the Fall Line (99%) than above (60%) and represented 34% of samples south and 11% north of the Fall Line (Table 1). *Reticulitermes*
*virginicus* was identified statewide from 318 sites (20%) and 122 counties (78%), with a slightly higher proportion of counties below (84%) than above (70%) the Fall line (Table 1). Statewide, *R. virginicus* was 11% of all samples (n = 494), with the proportion north of the Fall line (7%) more than doubled (15%) south of that geologic feature (Table 1). *Reticulitermes malletei* was identified from 73 counties (46%), collected at 323 sites (21%) with a much higher proportion of counties above (95%) than below (5%) the Fall Line (Table 1). *R. malletei* was represented statewide in 557 samples (13%) divided into 28% above and > 1% below the Fall Line (Table 1). *Reticulitermes hageni* was the least collected species identified from 18 counties (11%), 21 sites (1%), and 29 samples (> 1%), with all collections (100%) north of the Fall Line (Figure 2, Table 1).

### 3.2. Spatial Data Analysis

Hill Numbers (r = 0, 1, 2, ∞) support the aforementioned distributions because there were N0 = 5 species collected north the Fall Line (N1 = 3.55) compared to N0 = 4 south (N1 = 2.88) (Table 2). Hill Number (N2) Reciprocal Simpson’s index values likewise reflected the species distributions for above and below the Fall Line, N2 = 3 and N2 = 2.65, respectively (Table 2). Clark and Evans aggregation indices supported a regular distribution for Rf, Rn, and Rv across the state, while Rm and Rh had significantly clustered distributions in the northern part of the state (Table 3). Kernel Density Estimation [kde] maps illustrate the interplay between site and county distributions. Despite being distributed throughout the state, Rn was more prevalent in the south (Figure 3). Rv, despite its statewide distribution, was, proportionally, more frequently collected in the southern part of the state while Rm and Rh clustered north of the Fall Line (Figure 3).

## 4. Discussion

The survey identified five *Reticulitermes* species across a landmass encompassing ecological sub-regions representative of the southeastern United States east of the Appalachian Mountains into the mid-Atlantic states (Figure 1) [35]. The transition zone between the Piedmont and the Southern Coastal Plains, formed about 500 million years ago (the Fall Line), is a demarcation of an ancient shoreline bordering a shallow sea that started receding 20 mya before taking its current position about 5 mya [36]. An examination of species proportions across Georgia shows widespread, regular distributions for Rf, Rn, and Rv, while Rh and Rm were found predominantly north of the Fall Line. This biogeographic distribution, similar to *Loxoceles reclusa* in the same state [49], raises interesting questions about the evolution and dispersal abilities of these species that should inspire future research projects. 

A literature search of termite field studies in the southeastern USA published between 2012–2022 turned up 6 of 17 that mention either *R. malletei* or *R. nelsonae* (Appendix A). The prevalence of *R. nelsonae* in Georgia, collected in 81% or 128 counties statewide, including 99% of counties and 34% of samples south of the Fall Line (Table 1), provides solid evidence it has been underrepresented in the literature. Strict deference to literature published before the descriptions of *R. malletei* and *R. nelsonae* can skew *Reticulitermes* distribution estimates and underestimate the biodiversity of this genus across a wider geographic area. The only published dichotomous key to all five eastern *Reticulitermes* species [27] states that Rn has a southerly distribution, which the present data clearly disputes while highlighting the possibility that much of the Rh reported from the southeastern USA are Rn [30,50,51,52,53,54,55]. We present three lines of evidence, morphological, behavioral (flight phenology), and genetic, to rationalize a re-examination of the species distributions in the United States and present a path toward a concerted global effort to identify *Reticulitermes* biodiversity. 

Characteristic of the genus, a comparison of morphological measurements from descriptions of Rh and Rn reveals no quantifiable characters that clearly separate the diagnostic castes, although Rn alates are generally smaller (7.08 ± 0.29 mm) than Rh (7.81 ± 0.31 mm) [27,56]. The same ‘phenotypic overlap’ is revealed with qualitative characters where Rh “winged” forms were described as “Pale yellowish brown”, similar to the Rn “alate” described as “Body pale brown” [27,56]. There is, however, a significant behavioral character, flight phenology, that separates Rn from Rh. The description of Rh includes an alate Type Specimen collected north of the Fall Line, in Falls Church, Virginia, and the description states “In the vicinity of Washington (DC *sic*), *hageni* flies the later part of July or early in August” [56,57]. The description also contains the following entry: “Occurs from Florida (Jacksonville, April 29) to Maryland west to Illinois and Texas”, which has generally assumed to reference an ‘out-of-place’ collection of alates from FL in late April [56]. It is understandable prior to the description of Rn that passage in the original description provoked identifying ‘light-colored’ adult termites flying in Florida in springtime as Rh [24,55]. The original description of Rn reports adults collected in February, March, and May, while the present survey provided five Rn alate samples dated March and April [56]. All the ITA verified Rh (n = 17) in our archived assembly of alates (unpublished data) were collected in July–September. Rn flight phenology strongly suggests that the “R.h.” collected from springtime swarms in the United States should, regardless of precedence and in light of the more recent species description, be considered Rn. 

The molecular markers (COI, COII, or 16S sequence) used to identify the five focal *Reticulitermes* species provide an equally compelling argument for the prevalence of Rn in southeastern USA. We searched GenBank and included sequences attributed to the five species with the ITA reference sequence (Appendix A) to produce phylogenies (Figure 4). We found (186) COI GenBank accessions assigned to Rf, Rv, Rh, Rm, and Rn, and 185 aligned with their respective ITA-reference sequence except for one anomaly, collected in Florida, attributed to Rh that aligns with Rn (Appendix A). COII accessions from GenBank attributed to all Rf (n = 558), Rn (4), and Rm (8) agreed with the ITA references, including an Rn sample collected in Louisiana and an Rm from Mississippi [58], which extends the western range of both species. The southern range of Rn was illuminated by three samples from Florida, including one of our ITA references [27,58]. There were 29 COII accessions in GenBank attributed to Rv, of which 21 agreed with the reference sequence while seven direct submissions grouped with Rn and one with Rm references (Figure 4, Appendix A). There were nine Rh Genbank accessions and five grouped with the reference sequences, while anomalies included previously published incongruous Rh accessions that, unfortunately, have been used in studies published in the past decade (Appendix A). There were four ‘Rh’ COII accessions, including one collected in Arkansas and seven ‘Rh’ 16S accessions, with one collected in Brazos, TX, that grouped with the Rn references, further extending the potential western distribution of Rn (Appendix A). We posit, given the aforementioned morphological, behavioral, and genetic evidence and because we collected Rh infrequently (< 1% of > 4000 samples) and only from North Georgia, that the historic range of Rh has been overestimated and that of Rn underestimated.

*Reticulitermes malletei*, originally described from specimens collected near Athens, Georgia, in 1986 using chemical characters later clarified with morphological and genetic characters in 2007, has been reported from AL, MD, NC, SC, DE, GA, and MS [19,27,58,59,60,61,62]. The distribution of Rm from this survey (Figure 2), almost exclusively in north Georgia (Figure 1), coincides with its’ published range along the eastern coastal states, yet when combined with a GenBank accession from Indiana, our reference sequence from Mississippi and a GenBank accession from the same state [59] (Appendix A) begs further investigation of their western distribution. The qualitative phenotypic character “dark wings” used in dichotomous keys for both Rm and Rf likely provoked misidentification affecting historic range estimates for Rm [20,63,64]. Quantifiable morphological characters for Rm and Rf overlap for the soldier caste while for the adult ablw measurements show Rm to be, in general, smaller (8.23 ± 0.39 mm) than Rf (8.97 ± 0.40 mm) [20]. The adult Rm from this survey were collected in May–June and Rf from February–May, indicating a temporal overlap that needs further clarification. 

These data represent the largest survey of WOG over a contiguous land area, 153,900 km^2^, ever conducted for subterranean termites in the United States, but we caveat it should not be considered definitive. This research program, for example, has been located in Clarke County, Georgia, for 20 years, and we have collected Rh and Rv on several occasions, although the present survey did not collect either species in that county (Figure 2). This survey of wildland sites also failed to collect another invasive subterranean termite, *Coptotermes formosanus,* despite our program having identified *C. formosanus* from the built environment in Atlanta, Columbus, Savannah, Brunswick, Hinesville, and Thomasville, Georgia, in addition to a wildland site on Cumberland Island. Notwithstanding that proviso, the collection data from this extensive survey, when combined with examination of GenBank accessions and published literature, illuminated how *Reticulitermes* biodiversity in the eastern United States has been underestimated. The *Reticulitermes* literature in the USA continues to reference any number of approaches to species identification without heeding the taxonomic inconsistencies discussed 15 years ago [21].

The difficulties associated with identifying biologically cryptic insect complexes that display phenotypic plasticity make *Reticulitermes* an ideal candidate for species-specific molecular markers [65,66,67,68]. The monophyletic status of the genus makes it appropriate for a common set of ICZN-compliant, ITA-supported molecular marker(s) for species determination [69,70,71]. A common DNA barcode should interface with datasets of worldwide biodiversity and could set the foundation for broader discussions on a unified definition of what constitutes a species within this genus [72,73]. We suggest *Reticulitermes* surveys employ an ITA-validated mitochondrial DNA marker, specifically COI and/or COII sequence, as the species marker of choice given their widespread employment and reported advantages while acknowledging potential conflicts of linking phenotype with genotype [74,75]. We conducted a simple proof-of-concept test using the 55 COII reference sequences from this survey (Appendix A) with GenBank accessions attributed to *Reticulitermes* species from the Western USA, Europe, and Asia (Appendix A). All published species separated into strongly supported species-specific clades suggesting the utility of that mtDNA marker on a global scale (Figure 5). It is likely that the 16S sequence is too conserved to be useful across a global survey as there were 2 Rm 16S-haplotypes obtained from 22 samples collected across five states (estimated 1045 km linear distance), while we identified 15 Rm COII-haplotypes from 29 samples across five counties in north Georgia (estimated 60 km linear distance) [60,76]. In addition, attempts to corroborate 16S GenBank accessions for Rf, Rh, Rv, Rm, and Rn were thwarted by entries containing premature stop codons and ambiguities, leaving disproportionate gaps in the multiple sequence alignment needed for analysis [51,52,65,77]. Microsatellite markers or other electromorphs would be least preferred because they exhibit a propensity toward size homoplasy that would require additional research to verify repeatability and corroborate with IZCN-validated species designations [78,79,80].

This survey illustrates that subterranean termite biodiversity has been underestimated in the eastern United States and adds voice to the need for a concerted effort to develop a worldwide database suitable for assisting a generic taxonomic revision of *Reticulitermes*. That species complex would benefit from a formal reorganization following the ICZN code to provide an appropriate interpretation of the taxonomic literature and help correct the current muddle associated with inappropriate species attributions in the literature and datasets such as GenBank. In the meantime, the research community is encouraged to employ ITA-validated molecular markers to appropriately address biologically relevant research. Delineating species-level subterranean termite biodiversity should inspire future research in a broad array of topics, including evolution, ecology, sympatry, resource partitioning, and meaningful, identification-based pest-status monitoring of these important ecosystem engineers.

## Figures and Tables

**Figure 1 insects-13-00565-f001:**
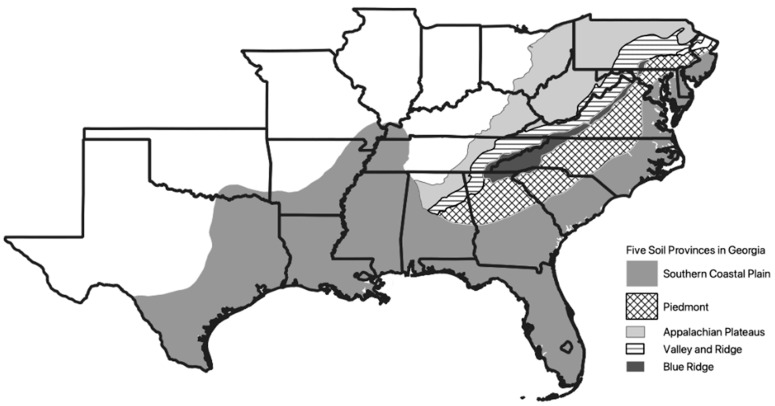
Map of the eastern United States outlining the 5 major soil provinces in Georgia that extend through other states in southeastern USA. Two soil provinces, the Piedmont and Southern Coastal Plain, occupy 90% of the landmass of Georgia and are demarked by the Fall Line that extends from Alabama to New Jersey.

**Figure 2 insects-13-00565-f002:**
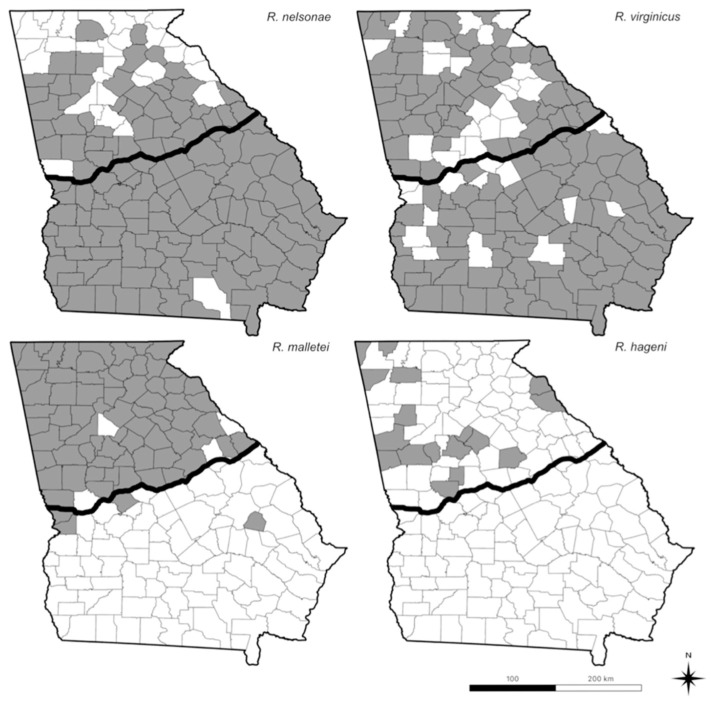
Distribution of four *Reticulitermes* species, out of P = 159 counties, in Georgia: upper left; *R. nelsonae* (P = 128); upper right; *R. virginicus* (P = 122); lower left; *R. malletei* (P = 73) and lower right; *R. hageni* (P = 18). Counties where designated species were found are shaded in dark gray for each map. A map of *R. flavipes* distribution was not included because it was found in every county. The dark line running east to west represents the position of the Fall line separating the Piedmont and Southern Coastal Plains soil provinces.

**Figure 3 insects-13-00565-f003:**
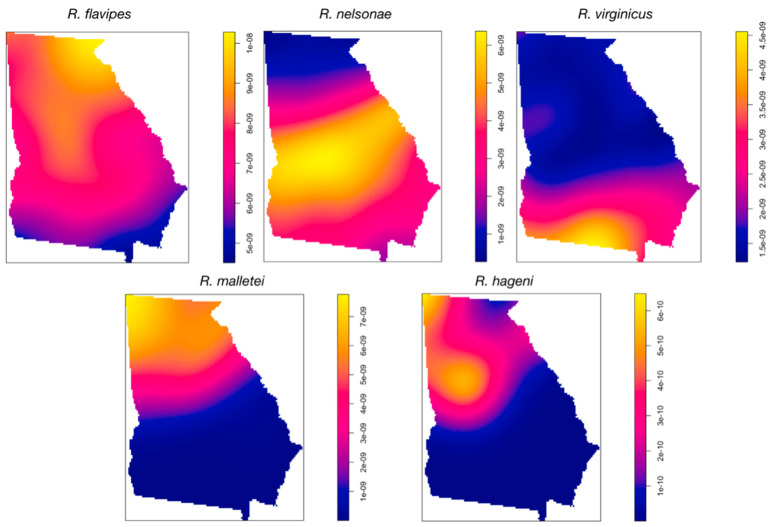
Kernel Density Estimation maps of *Reticulitermes* species distribution using site by species created in R. 4.0.5 (The R Foundation for Statistical Computing, Vienna, Austria). Areas of density based on distance between samples in km^2^, from high to low, are highlighted yellow to blue.

**Figure 4 insects-13-00565-f004:**
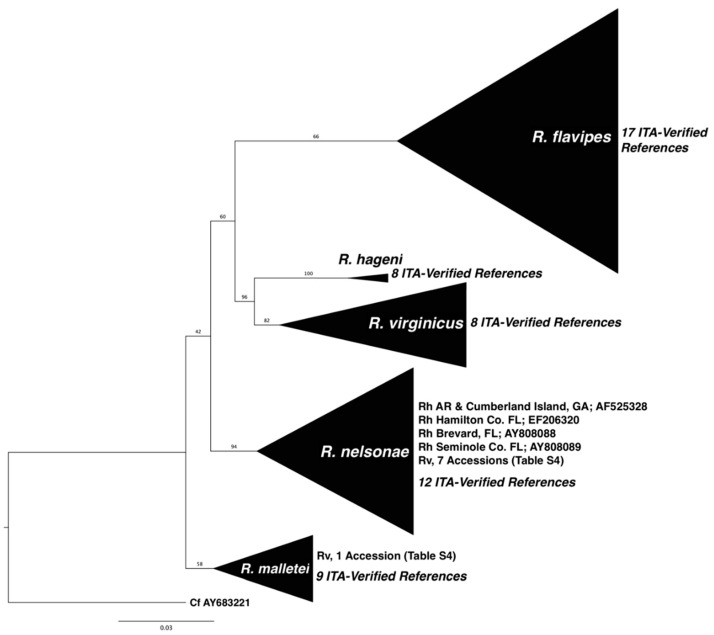
Collapsed Maximum-Likelihood GTR + G + I tree topology used to identify the survey samples to species, including 1186 full-length (685 bp) COII reads from the survey, highlighting the 55 full-length ITA-Verified references by species clade (Appendix A). A *Coptotermes formosanus* COII sequence (AY683221) was used as the extant group. Branch numbers represent percentage of posterior probability from 1000 UF Bootstrap iterations. The tree also included 3 full and 1 aligned partial reads from GenBank COII accessions shown by collection site and accession number reported as Rh that should be considered anomalous because they grouped with Rn (Appendix A). Full data available upon request from authors.

**Figure 5 insects-13-00565-f005:**
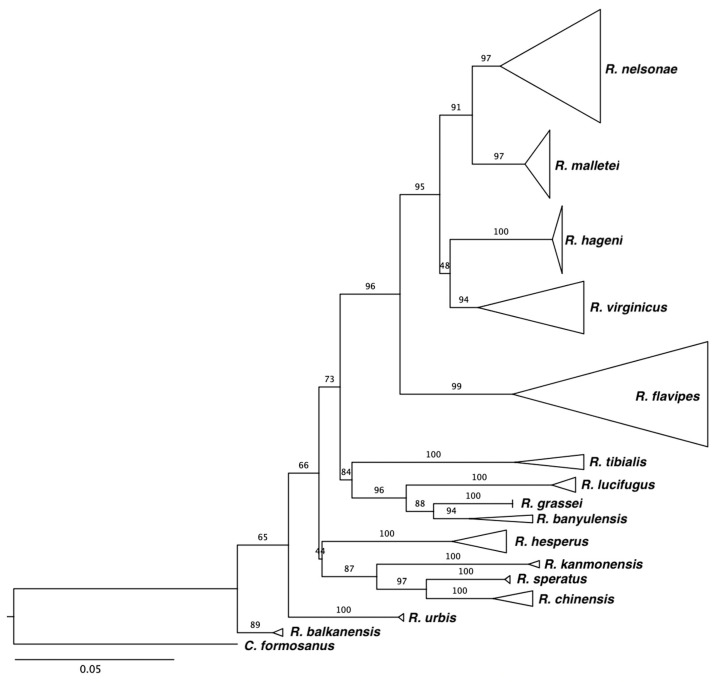
Collapsed Maximum-Likelihood GTR + G + I tree topology assembled using 55 full-length (685 bp) COII ITA-verified references for southeastern *Reticulitermes* (Appendix A) and 22 full or partial reads of the same gene retrieved from GenBank for 10 additional species across Europe, Asia, and the western United States (Appendix A). A *Coptotermes formosanus* sequence (AY683221) was used as the extant group. Strong node support from 1000 UF bootstrap iterations illustrates the global utility of this marker. Full data available upon request from authors.

**Table 1 insects-13-00565-t001:** The number and proportion of counties, sites, and samples by *Reticulitermes* species collected from the entire state as well as north and south of the Fall Line.

**Counties**						
	**Statewide**		**North of Fall Line**		**South of Fall Line**	
Species	# of counties (P = 159)	% of counties	# of counties (P = 73)	% of counties	# of counties (P = 86)	% of counties
Rf	159	100%	73	100%	86	100%
Rn	128	80.5%	43	58.9%	85	98.8%
Rm	73	45.9%	69	94.5%	4	4.7%
Rv	122	76.7%	50	68.5%	72	83.7%
Rh	18	14.1%	18	24.7%	0	0%
**Sites**						
	**Statewide**		**North of Fall Line**		**South of Fall Line**	
Species	# of sites (P = 1570)	% of sites	# of sites (P = 688)	% of sites	# of sites (P = 882)	% of sites
Rf	1131	72.0%	504	73.3%	627	71.1%
Rn	559	35.6%	130	18.9%	429	48.6%
Rm	323	20.6%	313	45.5%	10	1.1%
Rv	318	20.3%	97	14.1%	221	25.1%
Rh	21	1.3%	21	3.0%	0	0%
**Samples**						
	**Statewide**		**North of Fall Line**		**South of Fall Line**	
Species	# of samples (n = 4371)	% of samples	# of samples (n = 1975)	% of samples	# of samples (n = 2396)	% of samples
Rf	2260	51.7%	1036	52.5%	1224	51.1%
Rn	1031	23.6%	219	11.1%	812	33.9%
Rm	557	12.7%	546	27.7%	11	0.5%
Rv	494	11.3%	145	7.3%	349	14.6%
Rh	29	0.7%	29	1.5%	0	0%

**Table 2 insects-13-00565-t002:** Hill Numbers for *Reticulitermes* species collected by site from the entire state as well as north and south of the Fall Line.

Hill Numbers (r = 0, 1, 2, ∞)			
	Hill Number (N0) Number of Species	Hill Number (N1) Exponential Shannon (e^H^)	Hill Number (N2) Inverse Simpson (1/D)
**Entire State**	5	3.60	3.08
**North of Fall Line**	5	3.55	3.00
**South of Fall Line**	4	2.88	2.65

**Table 3 insects-13-00565-t003:** Clark and Evans Aggregation Index scores for *Reticulitermes* species distribution point patterns by site (R. 4.0.5, The R Foundation for Statistical Computing, Vienna, Austria).

Species	Clark and Evans R Score	*p*-Value
Rf	R = 1.19	<0.001
Rn	R = 1.05	<0.05
Rm	R = 0.71	<0.001
Rv	R = 0.99	0.79
Rh	R = 0.58	<0.001

## Data Availability

The data presented in this study are available on request from the corresponding author. A portion of the data is not publicly available due to time constraints in placing all 4000 samples into GenBank and pending publication of those same data in an examination of haplotypes.

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
