# Peer review of "Biodiversity and Distribution of Reticulitermes in the Southeastern USA"

_insects, 2022, doi:10.3390/insects13070565_

Round 1

Reviewer 1 Report

A well-written manuscript. I have only a small number of specific comments as follows.

Throughout the manuscript, the authors refer to each survey location as ‘locale', but it seems more appropriate to simply write ‘site'. The use of ‘locale(s)' gives the impression that the sites were selected on the basis of some environmental axis, whereas if the criteria are accessibility and geographical coverage, it would seem preferable to simply write 'site'. However, if 'locale' is appropriate as a common academic term in the authors' area of expertise and understandable for general readers, then it should be left as it is.

L 29 [identified from every county (n=159).] -> [identified from every of the 159 counties.]

The authors misuse the 'n'. Usually, 'n=##', is used to indicate the number of samples (the number of statistical trials), but the authors use 'n' in the manuscript to indicate the observed values. This is a confusing usage for the readers.

For example, in the Figure 2 legend (L196–199), the sample size (the number of statistical trials, i.e. the number of counties surveyed, here) is 159 for any termite species, so n=159 should be the case for the all species. However, the authors have written the observed values e.g. n=128 for R. nelsonae (i.e. the number of counties in which R. nelsonae was found, rather than the number of counties surveyed). This notation would mean that the number of counties where the distribution of R. nelsonae was surveyed was 128.

On the other hand, in Table 1, for example, the ‘(n=159)’ written in the table header correctly indicates the sample size (i.e. the number of counties surveyed), and the observations are written as values in the table.

Thus, there is a shaky notation, as if there is a fundamental lack of understanding of what 'n' is. Please unify the statistically correct usage of 'n' throughout the manuscript.

L 131, 132 [.2 µL] -> [0.2 µL]

L 186 [virginicus] -> in italics.

L 188–200 [the Fall Line]is spelt in different ways [the Fall line], [the fall line]. Please unify the notation.

L 198 The map of R. flavipes distribution is required. Whether 100% or 0%, a figure should be presented showing data on which of the counties surveyed were found, as with other species. In addition, the Figure 2 legend needs to be completely rewritten for reasons related to the use of n mentioned above. The meaning of the figure's paint-out should also be written, as it is not explained.

Figure 3 Lack of explanation of the colors scale. Does it represent population density? If so, what is the unit?

Table3 The p-value less than 0.001 should be reported as >0.001. In addition, reporting more than 3 significant digits does not give any useful information you know. Similarly, the significant digits of the statistic (the score) should be rounded.

L 234 [(n=128)]-> No. The sample size is 159.

L248, L308, [7.08-mm +0.29] -> What is ‘-mm’. 7.08±0.29 mm (mean±SD)? In L308, the unit is missing.

L 254 [hageni] -> in italics.

L 258 [that that] -> that

Author Response

We would like to thank the reviewer for making the time to share their expertise and provide comments on the submission Insects- 1778400 Titled Biodiversity and Distribution of Reticulitermes in the southeastern USA. 

We have provided the comments received in Blue italicized font while our responses are in Black font.

Throughout the manuscript, the authors refer to each survey location as ‘locale', but it seems more appropriate to simply write ‘site'. The use of ‘locale(s)' gives the impression that the sites were selected on the basis of some environmental axis, whereas if the criteria are accessibility and geographical coverage, it would seem preferable to simply write 'site'. However, if 'locale' is appropriate as a common academic term in the authors' area of expertise and understandable for general readers, then it should be left as it is.

The authors believe that the words Site and Locale are interchangeable and we deferred to the reviewer’s request by changing all reference to locales to sites throughout the manuscript.

L 29 [identified from every county (n=159).] -> [identified from every of the 159 counties.]

The authors misuse the 'n'. Usually, 'n=##', is used to indicate the number of samples (the number of statistical trials), but the authors use 'n' in the manuscript to indicate the observed values. This is a confusing usage for the readers.

For example, in the Figure 2 legend (L196–199), the sample size (the number of statistical trials, i.e. the number of counties surveyed, here) is 159 for any termite species, so n=159 should be the case for the all species. However, the authors have written the observed values e.g. n=128 for R. nelsonae (i.e. the number of counties in which R. nelsonae was found, rather than the number of counties surveyed). This notation would mean that the number of counties where the distribution of R. nelsonae was surveyed was 128.

On the other hand, in Table 1, for example, the ‘(n=159)’ written in the table header correctly indicates the sample size (i.e. the number of counties surveyed), and the observations are written as values in the table.

Thus, there is a shaky notation, as if there is a fundamental lack of understanding of what 'n' is. Please unify the statistically correct usage of 'n' throughout the manuscript.

We appreciate the reviewer 1’s attention to detail and changed the notations in Figure 2 to indicate which numbers reference a specific Population and those that reference samples.

L 131, 132 [.2 µL] -> [0.2 µL]

We appreciate mention of this oversight; the correction was made to Line 140 in the revision.

L 186 [virginicus] -> in italics.

Thank you for noticing the correction was made to what is now Line 218 in the revision.

L 188–200 [the Fall Line]is spelt in different ways [the Fall line], [the fall line]. Please unify the notation.

This was an excellent suggestion and corrections were made to standardize the term as Fall Line throughout the revision.

L 198 The map of R. flavipes distribution is required. Whether 100% or 0%, a figure should be presented showing data on which of the counties surveyed were found, as with other species. In addition, the Figure 2 legend needs to be completely rewritten for reasons related to the use of n mentioned above. The meaning of the figure's paint-out should also be written, as it is not explained.

We appreciate this comment but respectfully disagree. There is no reason to add extra space to Figure 2 by adding a 5th map that can easily be ‘described’ in words.  We did not add a fully-shaded map for R. flavipes yet do agree that the Figure 2 caption needed mention of the not-shaded counties in each map and added that statement in Lines 238-240.

Figure 3 Lack of explanation of the colors scale. Does it represent population density? If so, what is the unit?

We agree that the caption for Figure 3 needed a better description of the color scale and what units were in those maps. The correction was made to Lines 27-429 in the revision.

Table3 The p-value less than 0.001 should be reported as >0.001. In addition, reporting more than 3 significant digits does not give any useful information you know. Similarly, the significant digits of the statistic (the score) should be rounded.

We are indebted to the reviewers’ attention to detail. Those points were corrected in Table 3 in the revision.

L 234 [(n=128)]-> No. The sample size is 159.

We agree and that point was corrected in the revision.

L248, L308, [7.08-mm +0.29] -> What is ‘-mm’. 7.08±0.29 mm (mean±SD)? In L308, the unit is missing.

We appreciate mention of this oversight; the correction was made to Line 474-475 in the revision.

L 254 [hageni] -> in italics.

The correction was made to Line 480 in the revision.

L 258 [that that] -> that

The correction was made to Line 484 in the revision.

Thank you again for your expertise and time spent providing valuable comments on the manuscript. 

Reviewer 2 Report

I am pleased to review your work entitled"Biodiversity and Distribution of Reticulitermes in the south eastern USA". The work looks perfect and comprehensive and I recommend acceptance in the current form 

Author Response

We would like to thank the reviewer for making the time to share their expertise and provide comments on the submission Insects- 1778400 Titled "Biodiversity and Distribution of Reticulitermes in the southeastern USA". 

We have provided the comments received in Blue italicized font while our responses are in Black font. 

I am pleased to review your work entitled "Biodiversity and Distribution of Reticulitermes in the southeastern USA". The work looks perfect and comprehensive and I recommend acceptance in the current form 

We very much appreciate the positive comments and thank the reviewer for their time and consideration. The first author would like to send a box of cookies to this reviewer if they would contact her.